# Decellularized rat brain extracellular matrix effectively induces the dopaminergic differentiation of human adipose-derived stem cells

**Hossein Faghih, Arash Javeri\*, Masoumeh Fakhr Taha\***

Department of Stem Cells and Regenerative medicine, Institute for Medical Biotechnology, National Institute of Genetic Engineering and Biotechnology (NIGEB), Tehran, Iran

\* mftaha@nigeb.ac.ir (MFT); arashj@nigeb.ac.ir (AJ)

## Abstract

The extracellular matrix (ECM) plays essential roles in regulating various aspects of nervous system development. The ECM can be obtained through decellularization techniques, which preserve the native structure of tissue while removing cells and genetic material. Despite recent advancements in decellularization methods, removing cells from brain tissue remains challenging due to its delicate mechanical structure. Moreover, previous studies have not specifically evaluated the impacts of decellularized brain ECM on dopaminergic specification of stem cells. Here, we decellularized rat brain sections using a combination of chemical and enzymatic factors. Successful decellularization of sections was confirmed by DAPI, Haematoxylin and Eosin and Masson's trichrome staining, laminin immunostaining, DNA content analysis, and scanning electron microscopy. The sections were then recellularized with human adipose tissue-derived stem cells (hADSCs) and subjected to dopaminergic differentiation using a combination of growth factors. Some ADSCs were also differentiated in gelatin-coated tissue culture plates, employing a conventional two-dimensional culture method. After 12 days, the differentiated cells in both conditions expressed certain neuronal markers, especially those related to dopaminergic differentiation. However, *GLI1*, *VMAT2*, *GIRK2*, and *TH* genes, as well as NEFL, FOXA2, LMX1A, and TH proteins were upregulated in the ADSCs differentiated on decellularized sections. Furthermore, *DDC* and *CALB* were exclusively expressed by the ADSCs on decellularized brain sections. Overall, our findings indicated the significance of decellularized brain ECM to serve as an effective bioscaffold for dopaminergic differentiation of hADSCs. This highlights the importance of decellularization techniques for the advancement of midbrain tissue engineering and regenerative medicine for Parkinson's disease in the future.

**Data availability statement:** Data that support the findings presented are incorporated within the paper. The original uncropped images of western blots and RT-PCR analyses have been included as Supporting Information files (S1 Data and S2 Data). Furthermore, additional data are available as Supporting Information (S3 Data).

**Funding:** This study was supported by the Iran National Science Foundation, Tehran, Iran [grant Number 97010238]. This research was supported by the Iran National Science Foundation, Tehran, Iran [grant Number 97010238]. The funders had no role in study design, data collection and analysis, decision to publish, or preparation of the manuscript. They only supported this research financially.

**Competing interests:** The authors declare that they have no conflict of interest.

## Introduction

Mammalian tissues and organs are primarily composed of cells and extracellular matrix (ECM), a highly organized three-dimensional network consisting of water, proteoglycans, and fibrous proteins like collagen, elastin, laminin, and fibronectin. While many ECM components are the same in different tissues, each tissue has a unique combination of proteins and polysaccharides [1,2]. The ECM provides mechanical support for the cells, maintains tissue homeostasis, retains a wide range of growth factors, and determines the physical and biochemical characteristics of each organ, such as its tensile and compressive strength, stability, and elasticity. Additionally, the ECM plays significant roles in tissue morphogenesis, differentiation, and regeneration [1,3–5]. Considering the important functions of the ECM, decellularized bioscaffolds obtained from different tissues and organs may provide excellent biomaterials for tissue engineering and regenerative medicine, primarily due to their superb biocompatibility.

Over the last decade, different methods, using chemical, physical, and enzymatic factors, have been developed for decellularization of various tissues, such as the heart [6,7], skin [8,9], lung [10,11], kidney [12,13], and brain [14–16]. Cellularity, density, lipid content, and thickness of tissues are determining factors in the selection of decellularization method. The primary goal of these methods is efficient removal of the cells and genetic material while retaining the ultrastructure and composition of the ECM during the decellularization process [17].

Despite significant advancements in decellularization techniques, removing cells from brain tissue remains difficult due to its loose mechanical structure and high fragility. The brain is the fattiest organ in the body, and its ECM primarily consists of hyaluronic acid, glycosaminoglycans, and proteoglycans, with only a small fraction of fibrous proteins [15,18,19]. Based on these features, several methods employing different combinations of chemical, physical, and enzymatic agents have been established specifically for decellularization of the brain tissue, and the resulting ECM has been used either in a solubilized form to be added into the culture medium [14,15,20–22] or in a solid state for three-dimensional reconstruction of nervous tissue [16,23,24]. Furthermore, some injectable bioscaffolds, such as conductive hydrogels [25] and nerve growth factor (NGF)-releasing cryogels, have been developed from brain ECM for regenerative applications [26].

Previous studies have demonstrated the importance of using decellularized brain ECM to improve differentiation and maturation of neurons, with various cell sources such as neuroblastoma cells [20], induced pluripotent stem cell (iPSC)-derived neurons [15], neural stem cells [16,27], and bone marrow-derived mesenchymal stem cells (BM-MSCs) [28] being utilized as the initial cells. However, no study has specifically examined the influence of brain ECM on the dopaminergic differentiation of stem cells in vitro.

Recently, we induced the differentiation of human adipose tissue-derived stem cells (hADSCs) to dopamine-secreting cells using a serum-free medium with B27 supplement and a dopaminergic inducing cocktail composed of sonic hedgehog (SHH), fibroblast growth factor-8 (FGF8), basic-fibroblast growth factor (bFGF), and

brain-derived neurotrophic factor (BDNF) [29]. The same induction medium was then used for dopaminergic differentiation of hADSCs on decellularized rat brain sections. Overall, our findings demonstrated the positive role of decellularized brain ECM in dopaminergic differentiation of hADSCs. This protocol could be valuable for brain tissue engineering and transplantation therapy of neurodegenerative disorders, like Parkinson's disease, in the future.

## Materials and methods

### Ethics and consent declarations

Adipose tissue samples were harvested from five healthy women during elective abdominoplasty at Erfan-Niayesh Hospital, Tehran, Iran (from 2021.6.01 to 2022.12.30). While adipose tissue is typically considered medical waste destined for disposal, all donors provided informed verbal consent before the surgical procedure. This consent process was witnessed and documented by both the operating surgeon and the attending anesthesiologist. The documentations confirming the verbal consents were submitted to the Bioethics Committee of the National Institute of Genetic Engineering and Biotechnology (NIGEB). This committee performed a comprehensive review of all study-related documents and approved the study, assigning it the approval number: IR.NIGEB.EC.1400.2.26.C, dated: 2021.5.16. The experiments were conducted in accordance with the Declaration of Helsinki.

To prepare decellularized brain ECM, five male Wild-type Wistar rats were housed in the Central Animal House (NIGEB) under standard conditions. All surgeries were conducted under deep anesthesia to minimize suffering. The protocols were reviewed and approved by the Bioethics Committee of the National Institute of Genetic Engineering and Biotechnology in accordance with the guidelines and regulations for care and use of laboratory animals, with the approval number: IR.NIGEB.EC.1400.2.26.B, dated: 2021.5.16.

### Isolation and characterization of human ADSCs

Adipose tissue samples were harvested from 40 to 50 years old healthy women (n = 5), and ADSCs were isolated and characterized as described previously [29,30]. In brief, adipose tissue was minced and digested using 2 mg/ml collagenase I in phosphate-buffered saline (PBS) containing 2% bovine serum albumin (BSA). After centrifugation, the stromal vascular fraction (SVF) was cultured in a growth medium consisting of Dulbecco's Modified Eagle's Medium (DMEM), 20% fetal bovine serum (FBS), and 100 U/ml penicillin and 100 µg/ml streptomycin (all from Thermo Fisher Scientific, Massachusetts, USA). The culture medium was changed every 2 days, and the cells were passaged after reaching 80–90% confluence.

For characterization, third-passaged ADSCs were detached using 0.25% trypsin/ ethylenediaminetetraacetic acid (EDTA), fixed in 70% cold ethanol and immunostained with the primary antibodies against CD73, CD90, CD105, and CD45 proteins (all from Abcam, Cambridge, UK), followed by fluorescein isothiocyanate (FITC)-conjugated goat anti-mouse IgG (Sigma-Aldrich, St. Louis, MO, USA). Some cells stained with FITC-conjugated goat anti-mouse IgG were considered as negative controls. The cells were examined using a BD FACSCaliburTM (BD Biosciences, Franklin Lakes, New Jersey, USA) and were analyzed using FlowJo 7.6.1 (TreeStar Inc., Ashland, OR, USA).

To evaluate the multipotential differentiation capability of the ADSCs, the third-passaged cells were cultured in adipocyte induction medium containing DMEM, 10% FBS, 1 µM dexamethasone, 100 µM indomethacin, 5 µg/ml insulin, and 500 µM isobutylmethylxanthine (IBMX) or osteocyte induction medium containing DMEM, 10% FBS, $10^{-8}$ M dexamethasone, 10 mM glycerol phosphate, 3.7 g/l sodium bicarbonate, and 0.05 g/l ascorbic acid (all from Sigma-Aldrich) for three weeks. After that, Oil Red O staining was used to identify adipogenic differentiation, and Alizarin Red S staining was used for detection of osteogenic differentiation.

### Decellularization of rat brain sections

Five Wild-type Wistar rats weighing approximately 150–200 g were obtained from the Archives of Razi Institute, Iran, and were housed in the Central Animal House (NIGEB) under standard conditions. The rats were anesthetized using an

intraperitoneal injection of 100 mg/kg ketamine and 10 mg/kg xylazine, followed by trans-cardiac perfusion with 0.9% NaCl. The brains were isolated, rinsed in PBS, and sliced into 2–3 mm-thick coronal sections after removing the cerebellum and brainstem. Then, the sections were transferred into the wells of 24-well tissue culture plates and decellularized using a protocol with two consecutive cycles. During the first cycle, the sections were incubated for 2 hours in deionized water (dH$_2$O), 16 h in 4% sodium deoxycholate (SDC, diluted in dH$_2$O), 15 min in PBS, 30 min in dH$_2$O, 90 min in 1% Triton X-100 (diluted in dH$_2$O), 15 min in PBS, 30 min in dH$_2$O, 30 min in 0.02% trypsin/0.05% EDTA, 15 min in PBS, 30 min in dH$_2$O and 90 min in 40 kU/ml DNase I. During the second cycle, the sections were incubated for 16 h in 4% SDC, 15 min in PBS, 30 min in dH$_2$O, 60 min in 2% Triton X-100, 15 min in PBS, 30 min in dH$_2$O, 60 min in 40 kU/ml DNase I, 15 min in PBS and 30 min in dH$_2$O. All decellularization steps were performed in a shaker incubator (70–100 rpm) at room temperature. All solutions contained 200 U/ml penicillin, 200 mg/ml streptomycin, and 0.2 µg/ml amphotericin B.

### Evaluation of decellularized rat brain sections

To examine the successful removal of the cells and retention of the ECM in decellularized rat brain sections, we used the following techniques:

### Staining the nuclei

Some native and decellularized rat brain sections were fixed using 10% neutral buffered formalin, stained with 1 µg/ml 4`,6-diamino-2-phenylindole (DAPI), and visualized using a fluorescent microscope (Nikon, Eclipse TE 2000U microscope, JAPAN).

### Histological analysis

Some native and decellularized sections were fixed in 10% formalin, embedded in paraffin, sectioned by microtome, and stained with hematoxylin and eosin (H&E) or Masson's trichrome.

### DNA content

Genomic DNA was extracted from 1 mg of lyophilized native and decellularized rat brain sections using a digestion buffer containing NaCl (10 mM), Tris (10 mM, pH = 8), EDTA (10 mM, pH = 8), 0.5% sodium dodecyl sulfate (SDS) and 250 ng/ml proteinase K followed by DNA precipitation using 3 M sodium acetate and ice-cold absolute ethanol. The DNA pellet was resuspended in 50 µL nuclease-free water, and DNA concentration was determined by a NanoDrop 2000 (Thermo Scientific, USA). Statistical analysis was performed by GraphPad Prism 8 (GraphPad Software Inc.), and the significance of differences was assessed through an Unpaired t-test.

Polymerase chain reaction (PCR) was carried out using a PCR Master Mix (Ampliqon A/S, Denmark) and specific primers for SRY-box transcription factor 2 (*SOX2*) and hypoxanthine phosphoribosyltransferase 1 (*HPRT*) according to the manufacturer's instructions. PCR products were visualized using 2% agarose gel electrophoresis.

### Laminin immunostaining

The paraffin-embedded sections of native and decellularized rat brain sections were deparaffinized with xylene, effectively cleared with absolute ethanol, and hydrated using descending degrees of alcohol. To retrieve antigens, samples were incubated for 30 minutes in citrate buffer at 97 °C. Triton X-100 and 10% goat serum were used to permeabilize the cell membrane and block the non-specific reaction, respectively. The samples were then incubated with the primary antibody against laminin (Anti-Laminin antibody; Abcam, ab11575), the secondary antibody (Biotinylated Goat Anti-Rabbit secondary antibody; Abcam, ab64261), streptavidin-horseradish peroxidase (HRP), and 3,3'-diaminobenzidine (DAB) substrate. A fluorescence microscope (Germany-AXIOM) was used for observation and imaging.

## Scanning electron microscopy (SEM)

Some native and decellularized brain sections were fixed using 2.5% glutaraldehyde for 4 h, dehydrated by increasing degrees of ethanol, followed by freeze-drying. The samples were then coated in a sputter coater with a thin layer of gold, and images were taken using a scanning electron microscope (TESCAN GROUP, a.s., Kohoutovice, Czech Republic).

## Recellularization of the rat brain sections

Third to fifth-passaged ADSCs were seeded on the decellularized rat brain sections at a concentration of $5 \times 10^4$ cells/ml and cultured in growth medium containing DMEM, 20% FBS, 100 U/ml penicillin and 100 µg/ml streptomycin to allow attachment of the cells to the brain ECM. After 24 h, recellularization of the brain sections was evaluated using DAPI staining.

## Dopaminergic differentiation of the ADSCs on decellularized rat brain sections

For dopaminergic differentiation, third to fifth-passaged ADSCs were seeded on the decellularized rat brain sections at a concentration of $5 \times 10^4$ cells/ml and allowed to adhere to the ECM. After 24 h, growth medium was changed to dopaminergic induction medium consisting of Neurobasal™ Medium (Thermo Fisher Scientific), 100 µg/ml penicillin and 100 U/ml streptomycin, L-glutamine (both from Thermo Fisher Scientific), 0.25X B27, 250 ng/ml SHH, 100 ng/ml FGF8 and 50 ng/ml bFGF. From day 9 of differentiation, 50 ng/ml BDNF (all factors from Sigma-Aldrich) was also added. Differentiation was performed over 12 days. Every three days, half of the differentiation medium was renewed.

A cohort of ADSCs was plated on 0.1% gelatin-coated tissue culture plates, employing a conventional two-dimensional culture method. The cells were plated at a density of $5 \times 10^4$ cells/ml and were differentiated in the absence or presence of the same dopaminergic induction medium.

## RT-PCR and qPCR

Total RNA was extracted using the High Pure RNA Isolation Kit (Roche Diagnostics GmbH, Mannheim, Germany). 1 µg of each sample's total RNA was reverse transcribed to cDNA using Hyperscript RT Master Mix (GeneAll, Korea). The synthesized cDNAs were used as templates for PCR using a PCR Master Mix (Ampliqon A/S, Denmark) along with specific primers outlined in Table 1. PCR products were observed through 2% agarose gel electrophoresis.

Quantitative real-time PCR (qPCR) reactions were performed using RealQ-PCR 2x Master Mix (Ampliqon A/S, Denmark) on a Rotor-Gene™ 6000 (Corbett Research, NSW, Australia) real-time PCR machine. Relative quantification of gene expression between different groups was performed using REST 2009 software (Relative Expression Software Tool, Qiagen GmbH, Hilden, Germany). Four replicates of each sample were included in the qPCR experiments, with β2-microglobulin (*B2M*) and glyceraldehyde-3-phosphate dehydrogenase (*GAPDH*) serving as reference internal controls.

## Immunocytochemistry and immunohistochemistry

For immunocytochemistry, the cells were fixed using 4% paraformaldehyde, permeabilized with 0.5% Triton X-100 (Sigma-Aldrich), blocked using 10% goat serum (Gibco), and incubated with monoclonal antibodies against tyrosine hydroxylase (TH) and anti-beta-tubulin III (TUJ1) proteins (both antibodies from Sigma-Aldrich). Anti-mouse Phycoerythrin (PE)-conjugated IgG antibody was used as the secondary antibody, and DAPI was used for nuclear counterstaining. The cells were then examined using a fluorescence microscope (Nikon, Eclipse TE 2000U, Tokyo, Japan). For immunohistochemistry, recellularized brain sections were paraffin-embedded, sectioned, deparaffinized, hydrated, and then subjected to immunostaining against TH and TUJ1 proteins, following the same procedure as described for immunocytochemistry.

## Western blot analysis

The differentiated cells were lysed using ice-cold radioimmunoprecipitation assay (RIPA) lysis buffer [50 mM Tris-HCl (pH 7.4), 150 mM NaCl, 1 mM EDTA, 1 mM EGTA, 1% NP-40, 1 mM sodium orthovanadate (Na3VO4), 50 mM sodium

**Table 1. Primers used for RT-PCR and quantitative real-time PCR.**

| Gene | Forward | Reverse | Size | Accession |
|------|---------|---------|------|-----------|
| NSE | ACCTCAAGATGTCCCTCAGC | TCAGGACTGGGAGCAAAGATC | 186 | NM_006617 |
| NEFL | CTATGCAGGACACGATCAAC | TTCCAAGAGTTTCCTGTAAGC | 146 | NM_006158 |
| EN1 | GCTATCCTACTTATGGGCTCA | CTCGTTCTTCTTCTTCTTCAGC | 154 | NM_001426 |
| GLI1 | CAATGAGAAGCCGTATGTATGTA | GTAGAAATGGATGGTGCCCG | 162 | NM_005269 |
| NURR1 | ACTATTCCAGGTTCCAGGCGA | TATGCTAATCGAAGGACAAACAGT | 211 | NM_006186 |
| PITX3 | AGCACAGCGACTCAGAAAAGG | TCTTGAACCACACCCGCACG | 225 | NM_005029 |
| DDC | CTCATCCGATCAGGCACACT | GCAACCATAAAGAAAGGAATCAG | 165 | NM_001082971 |
| TH | CAGTTCTCGCAGGACATTGG | TTCACCTCCCCGTTCTGCTTA | 122 | NM_000360 |
| VMAT2 | GGTTTTGCAATTGGAATGGTGG | CCAGCAGAAGGACCTATAGC | 146 | NM_003054 |
| DAT | GCTTTCTCCTGTTCGTGGTC | CGTAGGCCAGTTTCTCTCGA | 192 | NM_001044 |
| GIRK2 | CTGACAGAATCCATGACTAACG | GTCTTTCCTCACGTACCTCTG | 171 | NM_002240 |
| CALB1 | AACTGAGGAGCTTAAGAACTTTC | CATCTCAGTTAATTCCAGCTTC | 145 | NM_004929 |
| B2M | TCCAGCGTACTCCAAAGATTCA | GTCAACTTCAATGTCGGATGGAT | 113 | NM_004048 |
| ACTB | CCTGGGCATGGAGTCCTGT | ATCTCCTTCTGCATCCTGTCG | 153 | NM_001101 |

fluoride (NaF)]. The RIPA buffer was supplemented with 1 mM phenylmethylsulfonyl fluoride (PMSF) and a protease inhibitor cocktail (Sigma Aldrich, Merck Group, St. Louis, Missouri, USA). The concentration of proteins was determined using the Bradford assay. Equal amounts of protein samples underwent SDS polyacrylamide gel electrophoresis, transferred to polyvinylidene fluoride (PVDF) membranes, and blocked using 5% skim milk (Sigma). The membranes were incubated with mouse monoclonal antibodies against β-actin (ACTB), TUJ1, NEFL, forkhead box protein A2 (FOXA2), and TH (all from Santa Cruz Biotechnology, Texas, USA) and a rabbit monoclonal antibody against LIM homeobox transcription factor 1 alpha (LMX1A) protein (Abcam) overnight at 4°C. Then, the membranes were exposed to a mouse IgG kappa binding protein (IgGκ BP)-HRP or mouse anti-rabbit IgG-HRP (both from Santa Cruz Biotechnology) secondary antibody at room temperature. The immunoreactive bands were detected using an electrochemiluminescence (ECL) kit (Cyto Matin Gene, Esfahan, Iran). ACTB served as the internal reference protein. Image analysis was conducted using Image Studio Lite Ver 5.2 (LI-COR Biosciences). Statistical analysis was performed using GraphPad Prism 8 (GraphPad Software Inc.), and the charts were generated accordingly. P values were obtained by an unpaired t-test with Welch's correction.

## Results

### Isolation and characterization of hADSCs

Flow cytometry assessment of the ADSCs for mesenchymal stem cell markers revealed that 98.5%, 99.3%, and 99.1% of the cells were positively stained with the antibodies against CD105, CD73 and CD90 proteins, respectively. Only 0.3% of the cells showed expression of CD45 as a hematopoietic marker (Fig 1A, S3 Data). To evaluate the multipotential differentiation capability of the ADSCs, the third-passaged cells (Fig 1B) were induced to undergo adipogenic and osteogenic differentiation. After three weeks of induction, lipid accumulation and calcium deposition were confirmed by Oil Red O staining (Fig 1C) and Alizarin Red S staining (Fig 1D), respectively.

### Decellularization of rat brain sections

In this study, a method was developed to decellularize rat brain sections using SDC, DNase I, Triton X-100, and weak trypsinization. As shown in Fig 2, the brain sections became progressively whiter and more transparent during each stage of decellularization. The successful decellularization of rat brain sections was confirmed by H&E and DAPI

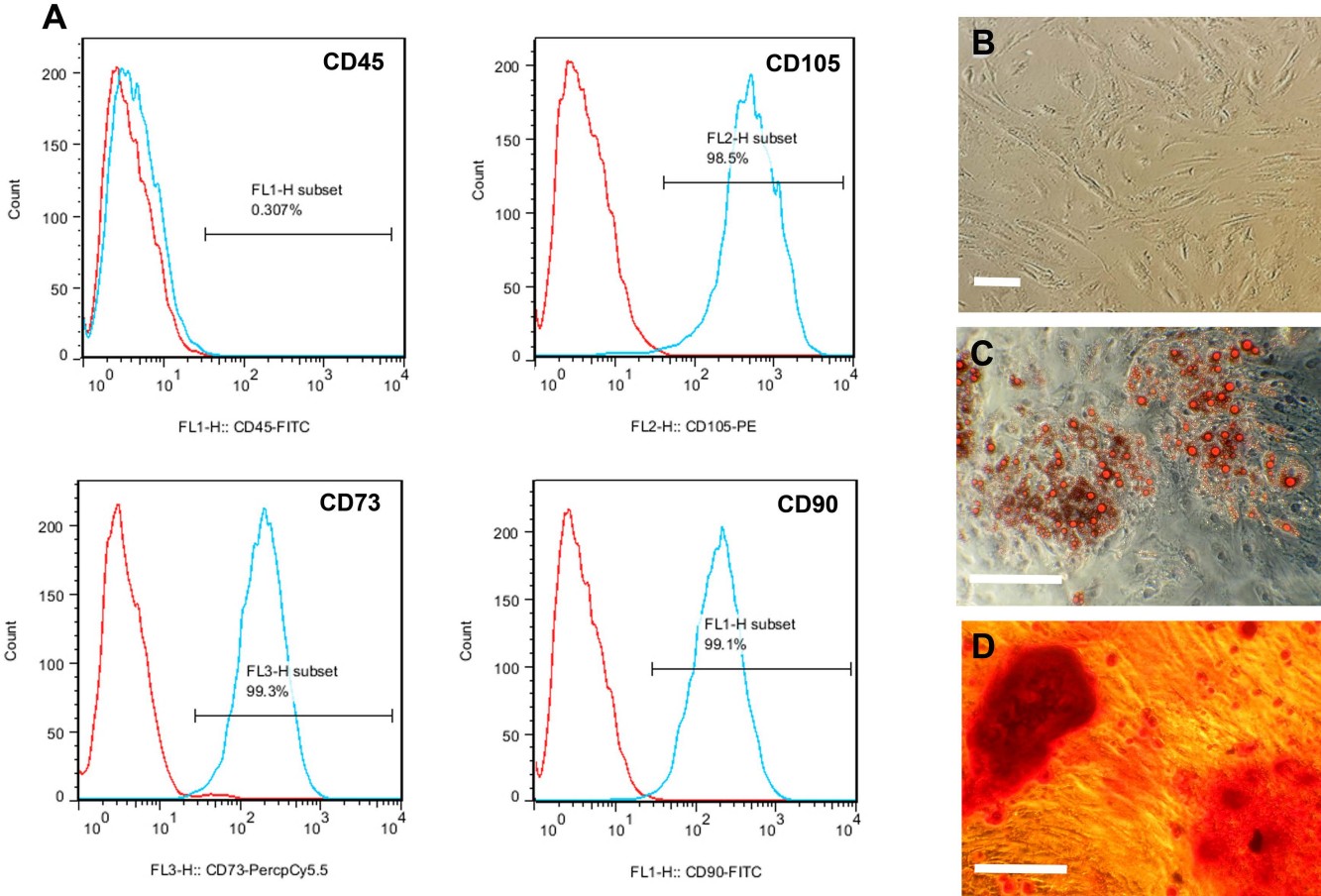

**Fig 1. Characterization of human ADSCs. (A)** Flow cytometry analysis of hematopoietic marker, CD45, and some mesenchymal stem cell markers, CD105, CD73, and CD90. **(B)** Fibroblast-like morphology of the third-passaged hADSCs. **(C)** Adipogenic differentiation of the ADSCs for three weeks; Lipid accumulation in the differentiated cells was confirmed by Oil Red O staining. **(D)** Osteogenic differentiation of the ADSCs for three weeks; calcium deposition in the differentiated cells was confirmed by Alizarin Red S staining.

staining, which revealed the absence of cells and nuclei in the decellularized rat brain sections (Fig 3A and 3C), whereas native brain sections showed the presence of cells and nuclei (Fig 3B and 3D). After decellularization, the amount of genomic DNA decreased significantly, reaching below 50 ng/mg dry weight of brain ECM (Fig 3E, S3 Data). Additionally, PCR analysis was performed for *SOX2* and *HPRT* on DNA samples extracted from both native and decellularized rat brain sections. Our results revealed the lack of PCR amplification for decellularized rat brain sections (Fig 3F, S1 Data).

Masson's trichrome staining and immunostaining against laminin confirmed the preservation of collagen fibres (Fig 4A and 4B, blue color) and laminin fibres (Fig 4C and 4D, brown color) in the decellularized rat brain sections. Moreover, SEM analysis revealed the presence of ECM fibers in the decellularized rat brain sections (Fig 4E-H).

### Recellularization of rat brain sections

After decellularization, rat brain sections were recellularized with hADSCs (Fig 5A). The existence of hADSCs on the brain sections was verified through DAPI staining (Fig 5B).

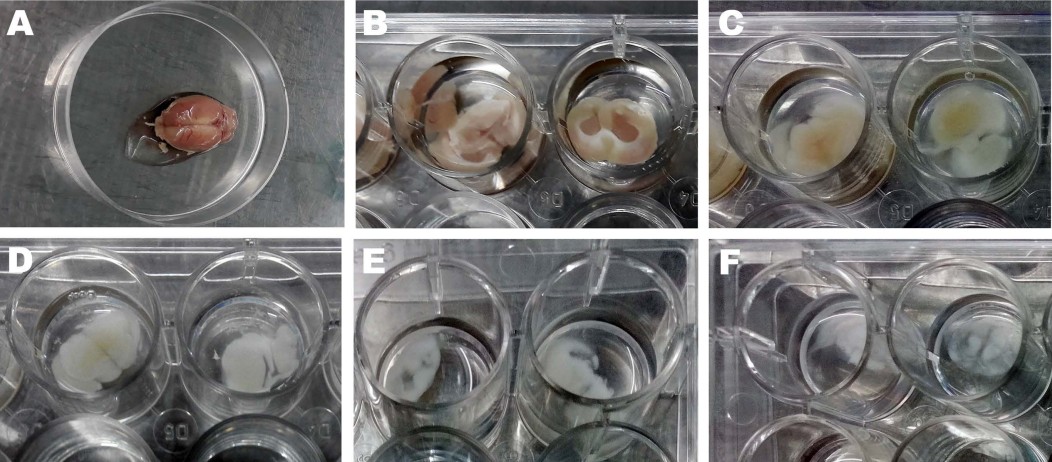

**Fig 2. Decellularization of rat brain sections. (A)** The rat brain was isolated after trans-cardiac perfusion with 0.9% NaCl. **(B)** 2-3 mm-thick rat brain sections were decellularized through sequential treatment with dH2O and 4% SDC **(C)**, Triton X-100 **(D)**, Trypsin-EDTA and DNase I **(E)**, and SDC, Triton X-100, and DNase I **(F)**.

## Dopaminergic differentiation of hADSCs on rat brain sections

Following dopaminergic differentiation of hADSCs on decellularized rat brain sections, as well as in gelatin-coated tissue culture plates, the expression of several markers associated with dopaminergic differentiation was evaluated. As indicated in Fig 5C, the differentiated cells in both conditions expressed Neuron-specific enolase (*NSE*), neurofilament light chain (*NEFL*), homeobox protein engrailed-1 (*EN1*), glioma-associated oncogene (*GLI1*), Nuclear receptor related 1 protein (*NURR1*), vesicular monoamine transporter (*VMAT2*), Tyrosine hydroxylase (*TH*) and G protein-coupled inward rectifying current potassium channel type 2 (*GIRK2*) genes (S1 Data). However, the expression of *GLI1*, *VMAT2*, *GIRK2,* and *TH* genes was significantly upregulated in the ADSCs differentiated on decellularized rat brain sections compared with those differentiated in gelatin-coated tissue culture plates (Fig 5D to 5G, S3 Data). Additionally, DOPA decarboxylase (*DDC*) and calbindin (*CALB*) genes were exclusively expressed by the ADSCs that were differentiated on decellularized rat brain sections (Fig 5C).

Next, it was demonstrated through immunostaining that a significant proportion of the differentiated ADSCs, both in gelatin-coated tissue culture plates (Fig 6A-F) and on decellularized rat brain sections (Fig 6G-L) expressed TUJ1 and TH proteins. However, as indicated by western blot analysis, the expression levels of TUJ1 (Fig 7A), NEFL (Fig 7B), FOXA2 (Fig 7C), LMX1A (Fig 7D) and TH (Fig 7E) proteins in the ADSCs differentiated on decellularized rat brain sections were higher than those differentiated in gelatin-coated tissue culture plates (S2 Data and S3 Data).

## Discussion

The ECM plays essential roles in regulating different aspects of nervous system development, including progenitor cell proliferation and differentiation, neural tissue morphogenesis, neurite outgrowth, synaptogenesis, synaptic signaling, and injury-related plasticity [31–33]. ECM signaling through certain molecules, like laminin, laminin-rich Matrigel and fibronectin, influences the developmental fate of embryonic stem cells (ESCs), resulting in the generation of highly enriched neural progenitor cells, neurite outgrowth, and functional specification of neurons, when applied in a dose-dependent manner at distinct stages of development [34–37]. Evidence also shows the regulating role of ECM in adult neurogenesis, synapse formation, and synaptic activity [31].

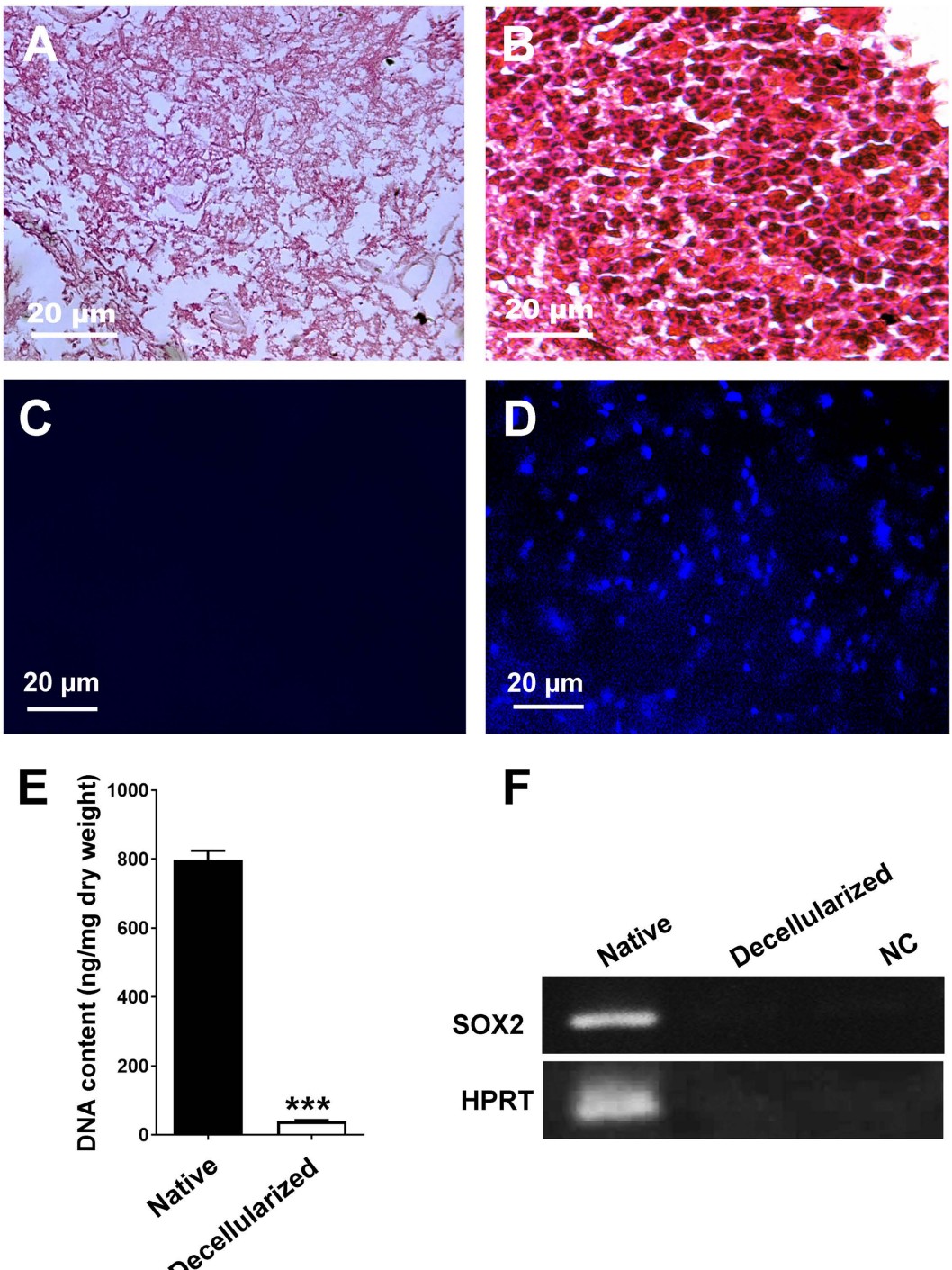

**Fig 3. Evaluation of decellularized rat brain sections. (A and B)** H&E staining of decellularized and native rat brain sections, respectively. **(C and D)** DAPI staining of decellularized and native rat brain sections, respectively. **(E)** DNA quantification, which revealed a decrease in DNA concentrations in the decellularized brain ECM to <50 ng per mg dry weight compared with the native brain tissue (n = 3, unpaired t test, ***:P < 0.001). **(F)** PCR analysis of the genomic DNA extracted from decellularized brain ECM and native brain tissue for *SOX2* and *HPRT* genes.

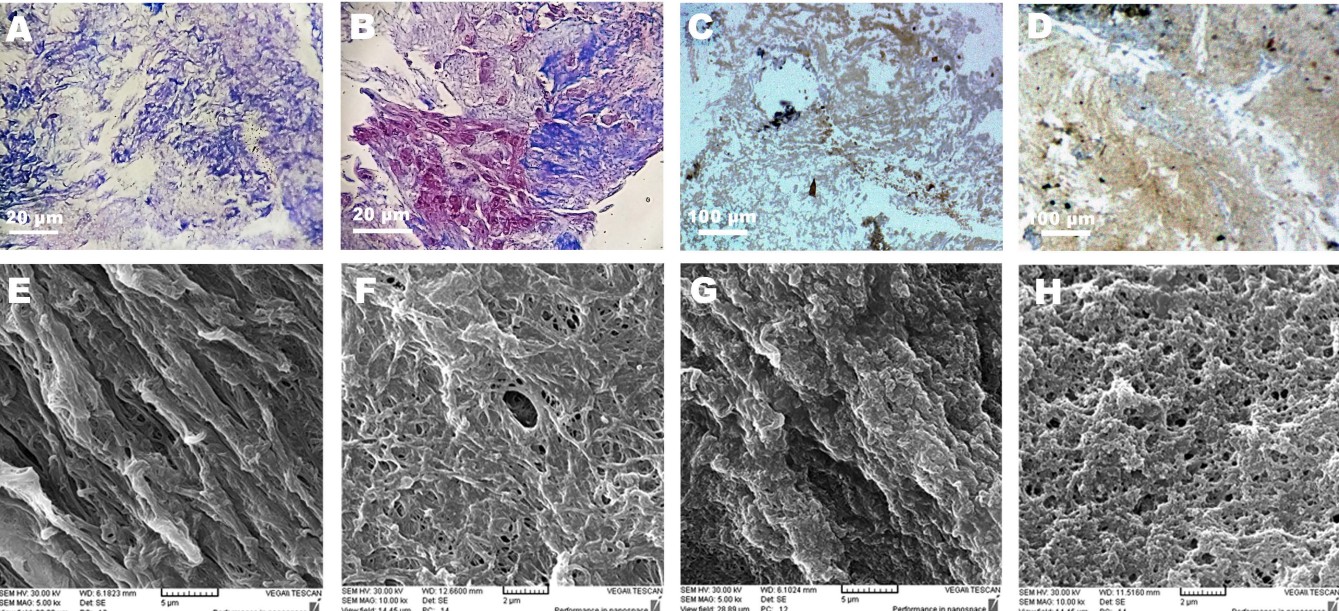

**Fig 4. Evaluation of decellularized rat brain sections. (A** and **B)** Masson's trichrome staining of decellularized and native rat brain sections, respectively, highlighting collagen fibres in blue. **(C** and **D)** Immunostaining against laminin in decellularized and native rat brain sections, respectively. Laminin has been shown to stain brown. **(E** and **F)** Scanning electron microscopy (SEM) images of decellularized rat brain sections and **(G** and **H)** the native sections, revealing the retention of ECM fibres in decellularized rat brain sections.

Until now, numerous studies have been conducted on the decellularization of various tissues, including the heart [6,7], skin [8,9], lung [12,13] and kidney [12,13]. However, research involving decellularization of brain tissue remains relatively limited [14–16]. This is possibly due to the loose mechanical structure and fragile nature of the brain tissue, making it more difficult to decellularize compared with other tissues.

In the present study, we developed a protocol for decellularization of rat brain sections through two consecutive cycles of treatments, involving 4% SDC, 1% Triton X-100, Trypsin-EDTA and DNase I in the first cycle, followed by 4% SDC, 2% Triton X-100 and DNase I in the second cycle, with immersion in dH2O and PBS after each step. This method is comparable to an approach described by De Waele et al. [16], albeit with notable distinctions. Firstly, we utilized a reduced concentration of Triton X-100, a most commonly used non-ionic detergent that disrupts the lipid-lipid and lipid-protein bonds, while preserving the protein–protein interactions [38]. Secondly, we performed a gentle Trypsinization at the first cycle of decellularization. Trypsin specifically acts on the bonds located on the C-side of arginine and lysine residues, while EDTA facilitates the disruption of cell–matrix interactions [39]. Since prolonged exposure to trypsin–EDTA can cause severe mechanical weakening of tissue due to changing the structure of matrix, degrading laminin, and removing glycosaminoglycans (GAGs) [40], we limited the concentrations to 0.02% trypsin and 0.05% EDTA for a duration of 30 minutes. Lastly, all decellularization steps were carried out in a shaker incubator at 70–100 rpm to prevent any damage to the scaffold. Overall, this method efficiently removed the cells and genetic materials from rat brain sections while retaining the ECM scaffold. To validate this, multiple assessments were performed on both native and decellularized rat brain sections.

H&E and DAPI staining revealed no visible cellular or nuclear components in the decellularized rat brain sections. After decellularization, the amount of genomic DNA decreased to under 50 ng/mg dry weight of brain ECM, which is believed to be non-immunogenic [17]. PCR amplification for *SOX2* and *HPRT* genes showed the absence of specific PCR products in the DNA samples extracted from decellularized rat brain sections. SEM analysis demonstrated the retention of ECM fibers

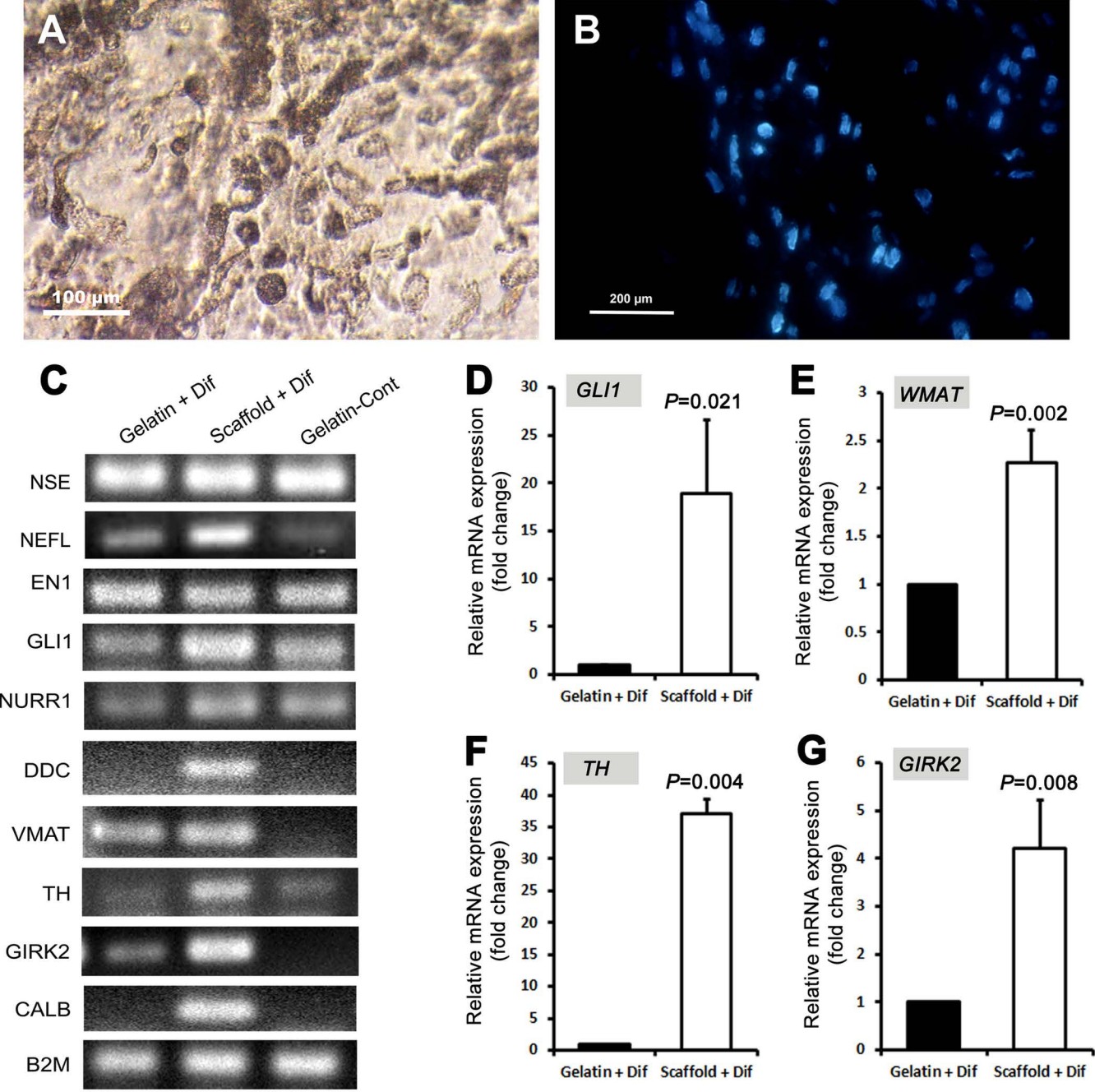

**Fig 5. Recellularization of decellularized rat brain sections and dopaminergic differentiation. (A)** The hADSCs were seeded onto the decellularized rat brain sections. **(B)** DAPI staining of the recellularized rat brain section. **(C)** RT-PCR and **(D-G)** qPCR analyses for the expression of some dopaminergic-related genes after dopaminergic differentiation of the ADSCs on decellularized rat brain sections or in gelatin-coated tissue culture plates. Gelatin+Dif: the ADSCs differentiated in gelatin-coated tissue culture plates. Scaffold+Dif: the ADSCs differentiated on decellularized rat brain sections. Gelatin-cont: the ADSCs cultured in gelatin-coated tissue culture plates with 10% FBS-containing medium. Significant differences are indicated by their respective P values (Pairwise Fixed Reallocation Randomization Test performed by REST 2009 software).

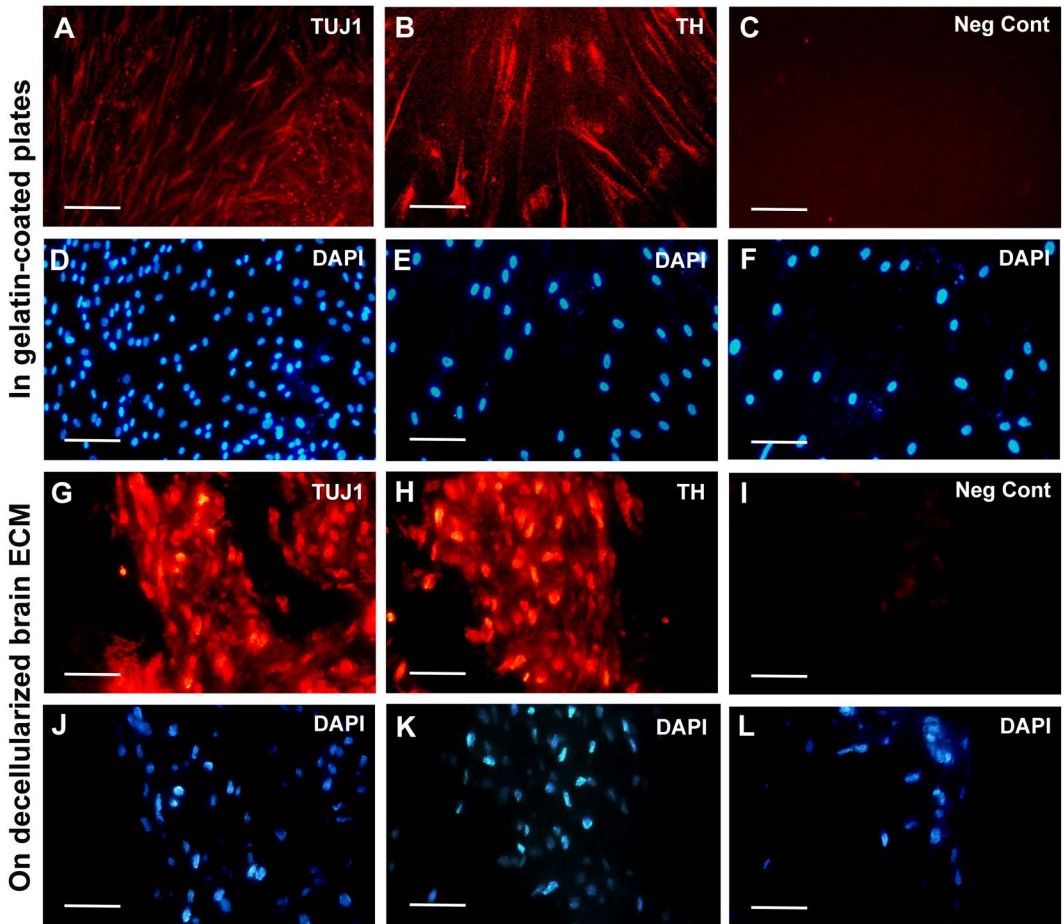

**Fig 6. Immunostaining of the differentiated ADSCs against TH and TUJ1 proteins. (A-C)** Immunocytochemical staining of the ADSCs differentiated in gelatin-coated tissue culture plates against TUJ1 and TH proteins, as well as a negative control (Neg Cont), where the primary antibody for the respective markers was omitted. **(D-F)** DAPI counterstaining of the nuclei in panels A to C, respectively. **(G-I)** Immunohistochemical staining of the ADSCs differentiated on decellularized rat brain sections against TUJ1 and TH proteins, as well as a negative control, where the primary antibody for the respective markers was omitted. **(J-L)** DAPI counterstaining of the nuclei in panels G to I, respectively.

in the decellularized brain sections. Additionally, Masson's trichrome staining and immunostaining against laminin fibers validated the retention of collagen and laminin fibers in the decellularized rat brain sections, respectively.

We recently directed the differentiation of hADSCs to dopamine-secreting cells using a specialized medium consisting of Neurobasal medium, B27 supplement, and a combination of growth factors, including SHH, bFGF, FGF-8, and BDNF [29]. In this investigation, we demonstrated through flow cytometry analysis that approximately 60% of hADSCs expressed TH protein after differentiation in 2D culture on gelatin-coated plates. The same induction medium was then used for the dopaminergic differentiation of hADSCs on decellularized rat brain sections. Furthermore, dopaminergic differentiation was conducted in standard gelatin-coated tissue culture plates, serving exclusively as a control to evaluate the differentiation rate in two- and three-dimensional conditions. The results from gene expression analysis indicated that both groups expressed *NSE, NEFL, NURR1, GLI1, EN1, TH, VMAT2*, and *GIRK2*, while *DDC* and *CALB* genes were exclusively expressed in the ADSCs differentiated on decellularized rat brain sections. As indicated previously, CALB is a neuroprotective factor that protects dopaminergic neurons from degeneration [41], and DDC plays a crucial role in

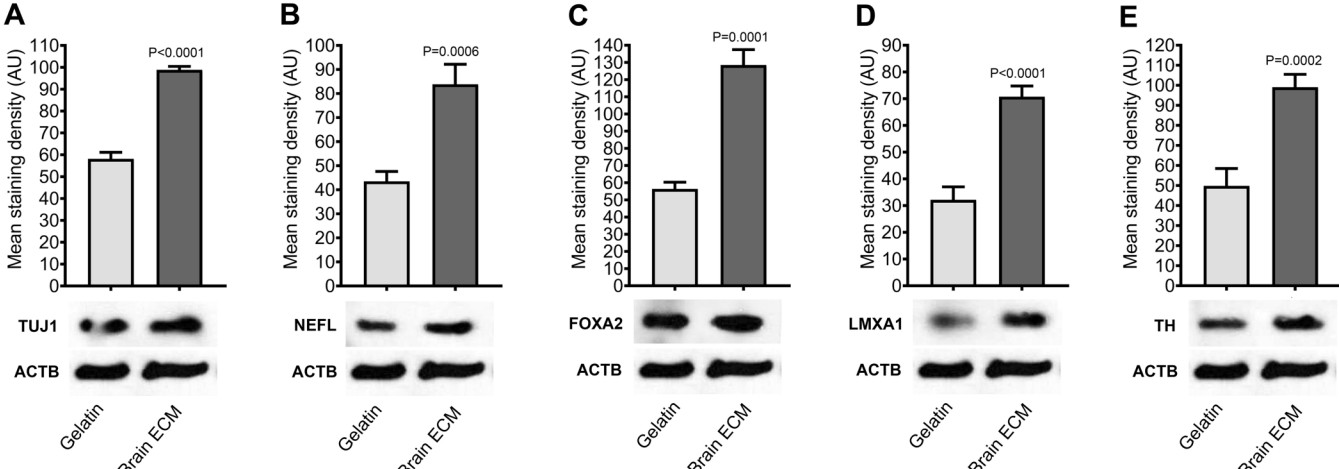

**Fig 7. Western blot analysis.** Western blot was performed for the expression of TUJ1 **(A)**, NEFL **(B)**, FOXA2 **(C)**, LMX1B **(D)**, and TH **(E)** proteins in the ADSCs that underwent differentiation in gelatin-coated tissue culture plates or on decellularized rat brain sections (n = 4). ACTB served as the internal reference protein. Image analysis was conducted using Image Studio Lite Ver 5.2 (LI-COR Biosciences). GraphPad Prism 8 (GraphPad Software Inc.) was used for statistical analysis and chart generation. *P* values were obtained by an unpaired t-test with Welch's correction.

the classical pathway of dopamine biosynthesis. This pathway begins with the hydroxylation of L-tyrosine by TH to yield L-dihydroxyphenylalanine (L-DOPA). Then, DOPA is decarboxylated to dopamine by DDC [42,43].

Based on qPCR analysis, *GLI1*, *VMAT2*, *GIRK2*, and *TH* were significantly upregulated in the ADSCs cultured on the brain ECM. Additionally, the ADSCs differentiated on decellularized brain sections exhibited a higher expression level of TUJ1, NEFL, FOXA2, LMX1B, and TH proteins compared with the cells differentiated on gelatin-coated surfaces. Collectively, these findings suggest the inductive role of decellularized brain ECM in the dopaminergic differentiation of hADSCs.

In our earlier research, we assessed the functional maturation of dopaminergic neurons generated from hADSC in 2D culture on gelatin-coated plates and demonstrated through high-performance liquid chromatography (HPLC) that a considerable amount of dopamine was released in response to KCl-induced depolarization [29]. In the present study, we did not examine the dopamine secretion by the differentiated cells on the brain ECM; however, based on our previous findings, we anticipate that the cells will secrete significant quantities of dopamine. Our marker expression analysis strongly implies a dopaminergic-like phenotype in the ADSC-derived neurons on the decellularized brain ECM. Nevertheless, a thorough functional validation remains a crucial future objective to ascertain neuronal maturity. Furthermore, another significant direction for future studies is to explore the differentiation of hADSCs on the decellularized brain ECM in the absence of the dopaminergic factors to determine whether the ECM alone can induce any neural differentiation.

As previously mentioned, the microenvironment surrounding the brain tissue significantly influences the development of neurons. This effect arises from the myriad dynamic interactions between the cells and the ECM [44]. Decellularized brain scaffolds effectively preserve the structural and functional complexity of the original tissue [45]. The fibrous components of the ECM, along with different growth factors which are preserved in decellularized brain ECM, play important roles in the development of neurons [14,46]. Furthermore, cultivating neural stem cells on the brain ECM and subsequently implanting them into the lesion site of the brain not only facilitates the transplantation of these exogenous cells but also induces the migration of endogenous cell populations into the damaged area [20]. On the other hand, transplantation of cell-free tissue-derived ECM presents a safer alternative, minimizing the risk of tumor formation from exogenous stem cells [47]. This approach provides a supportive microenvironment that encourages the proliferation, migration, and differentiation of endogenous neural stem cells, ultimately facilitating the replacement of lost neurons and regeneration of functional tissue

[48]. Collectively, these reports highlight the necessity of refining the methods of brain tissue decellularization and utilizing the decellularized bioscaffolds for the healing of central nervous system injuries, which formed the foundation of the present study. Our findings demonstrated clearly that an improved decellularization protocol involving two consecutive cycles yields a structurally preserved brain ECM that significantly promotes the dopaminergic differentiation of hADSCs, underscoring its potential for future application in brain tissue regeneration and healing of Parkinson's disease.

## Conclusion

In summary, we developed a technique to decellularize rat brain sections through two consecutive cycles of treatments involving 4% SDC, 1% Triton X-100, Trypsin-EDTA, and DNase I in the first cycle, followed by 4% SDC, 2% Triton X-100, and DNase I in the second cycle. This approach successfully removed the cells and genetic materials from the brain tissue, while preserving the ECM integrity. Furthermore, our findings indicated that decellularized rat brain sections can promote the dopaminergic differentiation of hADSCs. This method holds great potential for future applications in brain tissue engineering and transplantation therapy of neurodegenerative disorders, including Parkinson's disease.

## Supporting information

**S1 Data. Original, unprocessed images from RT-PCR analysis, corresponding to Fig 3 and Fig 5.**
(PDF)

**S2 Data. Original, unprocessed images of western blots, corresponding to Fig 7.**
(PDF)

**S3 Data. Raw data supporting the results illustrated in Fig 1, Fig 3, Fig 5, and Fig 7.**
(RAR)

## Acknowledgments

We thank the National Institute of Genetic Engineering and Biotechnology (NIGEB) for facilitating our research undertaken as part of the PhD thesis by the first author.

## Author contributions

**Conceptualization:** Masoumeh Taha.

**Data curation:** Hossein Faghih.

**Formal analysis:** Arash Javeri.

**Funding acquisition:** Masoumeh Taha.

**Investigation:** Hossein Faghih.

**Methodology:** Hossein Faghih, Arash Javeri.

**Supervision:** Arash Javeri, Masoumeh Taha.

**Validation:** Arash Javeri, Masoumeh Taha.

**Writing – original draft:** Hossein Faghih.

**Writing – review & editing:** Arash Javeri, Masoumeh Taha.

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
