## [Decision Letter · Decision Letter 0]

Dear Dr. Taha,

Thank you for submitting your manuscript to PLOS ONE. After careful consideration, we feel that it has merit but does not fully meet PLOS ONE’s publication criteria as it currently stands. Therefore, we invite you to submit a revised version of the manuscript that addresses the points raised during the review process.

We look forward to receiving your revised manuscript.

Kind regards,

Panayiotis Maghsoudlou

Academic Editor

PLOS ONE

2. In the ethics statement in the Methods, you have specified that verbal consent was obtained. Please provide additional details regarding how this consent was documented and witnessed, and state whether this was approved by the IRB

“This study was supported by the Iran National Science Foundation, Tehran, Iran [grant Number 97010238].”

4. In the online submission form, you indicated that [Insert text from online submission form here].

Reviewers' comments:

Reviewer's Responses to Questions

**Comments to the Author**

1. Is the manuscript technically sound, and do the data support the conclusions?

Reviewer #1: Yes

Reviewer #2: Yes

2. Has the statistical analysis been performed appropriately and rigorously?

Reviewer #1: Yes

Reviewer #2: Yes

3. Have the authors made all data underlying the findings in their manuscript fully available?

Reviewer #1: Yes

Reviewer #2: Yes

4. Is the manuscript presented in an intelligible fashion and written in standard English?

Reviewer #1: No

Reviewer #2: Yes

- Presentation Quality: The text and figures need refinement. The figures should be better described and integrated into the text, and the resolution of the Figures needs to be improved!

Reviewer #2: The manuscript "Decellularized Rat Brain Extracellular Matrix Effectively Induces Dopaminergic Differentiation of Human Adipose-Derived Stem Cells" presents data that supports its conclusions and includes adequate statistical analyses. It also states that the complete data will be made available. The quality of the English used in the manuscript is satisfactory. The manuscript meets all publication requirements and introduces valuable insights for advancing neuronal regeneration.

**Do you want your identity to be public for this peer review?** For information about this choice, including consent withdrawal, please see our Privacy Policy

Reviewer #1: No

Reviewer #2: No

---

## [Author Response · Author response to Decision Letter 1]

11 Feb 2025

Responses to reviewers 11 Feb 2025

Dear Editor,

We would like to thank the reviewers and the subject editor for the thoughtful critiques of our manuscript. We have revised the methods and materials section of the manuscript based on the queries and hope that now you find it suitable for publication in the PLOS ONE journal. Our responses to the questions are detailed on the following page.

Masoumeh Fakhr Taha, PhD

Department of Stem Cells and Regenerative Medicine

Institute for Medical Biotechnology

National Institute of Genetic Engineering and Biotechnology (NIGEB)

Tehran, Iran

Emails: mftaha@nigeb.ac.ir

Phone: +98 912 2999652

Thank you for your response. For complete transparency, in your Methods, please clearly state 1) how the verbal consent for the use of adipose tissue samples for research purposes was documented and witnessed, and 2) state whether this verbal consent procedure was approved by the IRB.

Answer:

1. In response to the first question, as outlined in the methods and materials section, adipose tissue samples were collected from healthy women undergoing elective abdominoplasty, at Erfan-Niayesh Hospital, Tehran, Iran (from 2021.6.01 to 2022.12.30). While adipose tissue is typically considered as pathological waste destined for disposal, the collection of these samples was conducted in conjunction with the operating surgeon. Prior to the surgical procedure, non-written (verbal) consent was obtained from the patients, a process that was witnessed and documented by both the operating surgeon and the attending anesthesiologist. All donors agreed to the use of their adipose tissue samples for research purposes. Further information regarding the identities of the medical professionals involved in this documentation can be provided upon request.

2. In regard to the second question, the documentations affirming the verbal consents were submitted to the Bioethics Committee of National Institute of Genetic Engineering and Biotechnology in order to obtain their approval. This committee conducted a thorough review of all study-related documents, including the consent forms, and subsequently approved the study, with the approval number: IR.NIGEB.EC.1400.2.26.C, dated: 2021.5.16.

Responses to reviewers 1 4 Feb 2025

Dear Editor,

We would like to thank the reviewers and the subject editor for the thoughtful critiques of our manuscript. We have revised the manuscript and hope that now you find it suitable for publication in the PLOS ONE journal. Our point-by-point responses to the comments are detailed on the following pages.

Masoumeh Fakhr Taha, PhD

Department of Stem Cells and Regenerative Medicine

Institute for Medical Biotechnology

National Institute of Genetic Engineering and Biotechnology (NIGEB)

Tehran, Iran

Emails: taha_tmu73@yahoo.com, mftaha@nigeb.ac.ir

Phone: +98 912 2999652

Answer: The manuscript has been thoroughly reviewed. We believe the manuscript meets PLOS ONE's style requirements.

2. In the ethics statement in the Methods, you have specified that verbal consent was obtained. Please provide additional details regarding how this consent was documented and witnessed, and state whether this was approved by the IRB.

Answer: As outlined in the methods and materials section, adipose tissue samples were collected during elective abdominoplasty surgery which is typically considered as pathological waste destined for disposal. Nevertheless, the collection of these samples was conducted in conjunction with the surgeon present in the operating room. Additionally, informed verbal consent was obtained from the donors by the surgeon before preparations for the surgical procedure. All donors agreed to the use of their adipose tissue samples for only research purposes. The experiments were conducted in accordance with the Declaration of Helsinki, and were reviewed and approved by the Bioethics Committee of National Institute of Genetic Engineering and Biotechnology with the approval number: IR.NIGEB.EC.1400.2.26.C, dated: 2021.5.16.

“This study was supported by the Iran National Science Foundation, Tehran, Iran [grant Number 97010238].”

Answer: The funders had no role in study design, data collection and analysis, decision to publish, or preparation of the manuscript. They only supported this research financially. We included this amended Role of Funder statement in our cover letter.

4. In the online submission form, you indicated that [Insert text from online submission form here].

Answer: Some data that support the findings presented in the manuscript are included within the manuscript itself, whereas additional data are provided as supplementary information (Figures S1 to S3).

Answer: The original uncropped images of western blots and RT-PCR analyses have been included as a supporting information file (Fig S1 and Fig S2).

Answer: The reference list has been thoroughly reviewed and is now both complete and accurate.

Comments to the Author

1. Is the manuscript technically sound, and do the data support the conclusions?

Reviewer #1: Yes

Reviewer #2: Yes

2. Has the statistical analysis been performed appropriately and rigorously?

Reviewer #1: Yes

Reviewer #2: Yes

3. Have the authors made all data underlying the findings in their manuscript fully available?

Reviewer #1: Yes

Reviewer #2: Yes

4. Is the manuscript presented in an intelligible fashion and written in standard English?

Reviewer #1: No

Reviewer #2: Yes

5. Review Comments to the Author

Reviewer #1: Presentation Quality: The text and figures need refinement. The figures should be better described and integrated into the text, and the resolution of the Figures needs to be improved!

Answer: We have revised the figures’ qualities and text and tried to improve them as much as possible. All figures are in TIFF format with 600 dpi resolution.

Reviewer #2: The manuscript "Decellularized Rat Brain Extracellular Matrix Effectively Induces Dopaminergic Differentiation of Human Adipose-Derived Stem Cells" presents data that supports its conclusions and includes adequate statistical analyses. It also states that the complete data will be made available. The quality of the English used in the manuscript is satisfactory. The manuscript meets all publication requirements and introduces valuable insights for advancing neuronal regeneration.

---

## [Decision Letter · Decision Letter 1]

Dear Dr. Taha,

Thank you for submitting your manuscript to PLOS ONE. After careful consideration, we feel that it has merit but does not fully meet PLOS ONE’s publication criteria as it currently stands. Therefore, we invite you to submit a revised version of the manuscript that addresses the points raised during the review process.

We look forward to receiving your revised manuscript.

Kind regards,

Panayiotis Maghsoudlou

Academic Editor

PLOS ONE

Reviewers' comments:

Reviewer's Responses to Questions

**Comments to the Author**

Reviewer #3: (No Response)

Reviewer #4: (No Response)

Reviewer #5: (No Response)

2. Is the manuscript technically sound, and do the data support the conclusions?

Reviewer #3: No

Reviewer #4: Yes

Reviewer #5: (No Response)

3. Has the statistical analysis been performed appropriately and rigorously?

Reviewer #3: Yes

Reviewer #4: Yes

Reviewer #5: (No Response)

4. Have the authors made all data underlying the findings in their manuscript fully available?

Reviewer #3: Yes

Reviewer #4: Yes

Reviewer #5: (No Response)

5. Is the manuscript presented in an intelligible fashion and written in standard English?

Reviewer #3: Yes

Reviewer #4: Yes

Reviewer #5: (No Response)

Reviewer #3: This study assesses the effect that decellularized rat brain slices have on the dopaminergic differentiation of human adipose-derived stem cells. However there are some serious flaws in the study design which preclude conclusions being drawn.

Firstly, a preference towards dopaminergic differentiation cannot be claimed as other markers were not assessed by PCR or ICC (e.g. other neuronal types and/or glia). To claim a preference for dopaminergic differentiation, this should be compared to the subpopulations of other cell types.

Secondly, regarding the ICC data, this is the most critical data to claim dopaminergic differentiation but the data is problematic. Firstly both TH and Tuj1 are cytoplasmic markers. However, the staining here does not have any neuronal morphology, looks distinctly nuclear, and there is almost complete overlap with DAPI, especially the TH staining. This brings into question the validity of this data. Additionally, the authors state that a "significant portion" of the cells expressed TH and TUJ1. However, not only is this not statistically analysed, it is not even quantified.

Thirdly, the control for the decellularized rat brain slices is not appropriate, or at least, other more appropriate controls should have been included. Why gelatin for example - this is not used in standard cell culture differentiation protocols and is not an ECM component highly expressed in brain tissue.

Finally, the data cannot be compared to existing literature as the time point for differentiation was not specified.

Reviewer #4: This is a general good original research. The authors must address these questions:

1- what are the main novelties of this research? It seems that a new method for decellularization has been conducted, but its efficacy and effectiveness has not been compared with common traditional approaches.

2- Since the control group of cells has received dopaminergic culture medium in two dimensional flask condition, the test group was cultured on 3D bio-scaffold. As that are substantial differences between two and three dimensional culture condition, the elevation of gene expression could not be exclusively related to the de-cellularized bio-scaffolds. More intricate and complex controlled condition is necessary to draw the claimed conclusions.

3- did the authors conduct any essay for cell sorting? Extraction of human adipose derived mesenchymal stem cells usually leads to a heterogeneous colony of the cells and scientific sorting approaches is necessary for compromising a homogeneous colony. Otherwise the results seem not to be repetitive.

4- For further and more complete investigation it is needed to assess the expression of neuron exclusive genes in both mRNA and protein levels.

Reviewer #5: (No Response)

**Do you want your identity to be public for this peer review?** For information about this choice, including consent withdrawal, please see our Privacy Policy

Reviewer #3: No

Reviewer #4: **Yes: ** Shahrokh Shojaei

Reviewer #5: No

---

## [Author Response · Author response to Decision Letter 2]

2 Jul 2025

Responses to reviewers 1 Jul 2025

Dear Editor,

We would like to thanks the reviewers and the subject editor for the thoughtful critiques of our manuscript. We have revised the manuscript and hope that now you find it suitable for publication in the PLOS ONE journal. Our point-by-point responses to the comments are detailed on the following pages.

Masoumeh Fakhr Taha, PhD

Department of Stem Cells and Regenerative Medicine

Institute for Medical Biotechnology

National Institute of Genetic Engineering and Biotechnology (NIGEB)

Tehran, Iran

Emails: taha_tmu73@yahoo.com, mftaha@nigeb.ac.ir

Phone: +98 912 2999652

Reviewer #3:

This study assesses the effect that decellularized rat brain slices have on the dopaminergic differentiation of human adipose-derived stem cells. However, there are some serious flaws in the study design which preclude conclusions being drawn.

Firstly, a preference towards dopaminergic differentiation cannot be claimed as other markers were not assessed by PCR or ICC (e.g. other neuronal types and/or glia). To claim a preference for dopaminergic differentiation, this should be compared to the subpopulations of other cell types.

Response: Thank you for your valuable comment. We acknowledge that a definitive claim of dopaminergic lineage preference requires comparative analysis with other neuronal and glial subtypes. However, in this study, we focused on dopaminergic differentiation of hADSCs and did not examine other types of neurons or glial cells, for the following reasons:

1. The combination of SHH, FGF8, bFGF, and BDNF is a known cocktail for dopaminergic differentiation of stem cells. Sonic hedgehog (SHH) and fibroblast growth factor (FGF8) are traditional factors that are used in dopaminergic differentiation [1]. A combination of SHH and FGF8 can induce dopaminergic neurons at ectopic locations in the rat embryo, while SHH alone can only induce dopaminergic neurons at the dorsal-ventral axis [2]. In previous studies, a combination of SHH, FGF8, and bFGF has been used for dopaminergic differentiation of mouse and human embryonic stem cells [3, 4]. In 2007, Trzaska and colleagues [5] used a combination of SHH, FGF8, and bFGF for dopaminergic differentiation of human bone marrow-derived mesenchymal stem cells. Also, in 2011, Trzaska et al. generated DA-producing cells from adult human bone marrow-derived mesenchymal stem cells using SHH, FGF8, and bFGF and showed that electrophysiological functional DA neurons could be achieved by further treatment with BDNF [6]. The same combination of growth factors was used later by Shall et al. [7] and Phonchai et al. [8] for dopaminergic differentiation of rat bone marrow-derived mesenchymal stem cells and human amniotic fluid mesenchymal stem cells, respectively. In 2018, our team used this cocktail (SHH, FGF8, bFGF, and BDNF) for dopaminergic differentiation of human ADSCs and generated dopamine-secreting cells, with the same method used in the current study [9].

2. We would like to clarify that in the study by Trzaska et al. (2007), who differentiated BM-derived MSCs using the same medium as we employed, there was no detection of glial markers, such as GFAP and Olig2, in the differentiated cells. This absence of glial marker expression supports the notion that the differentiation was not directed toward glial lineages. The findings from Trzaska et al. further support the dopaminergic tendency under similar induction conditions. Nevertheless, we acknowledge the importance of including broader lineage markers in future studies to confirm lineage specificity.

Secondly, regarding the ICC data, this is the most critical data to claim dopaminergic differentiation but the data is problematic. Firstly both TH and Tuj1 are cytoplasmic markers. However, the staining here does not have any neuronal morphology, looks distinctly nuclear, and there is almost complete overlap with DAPI, especially the TH staining. This brings into question the validity of this data. Additionally, the authors state that a "significant portion" of the cells expressed TH and TUJ1. However, not only is this not statistically analysed, it is not even quantified.

Response: Thank you for your critical observation regarding ICC images. We acknowledge the concerns raised about the staining pattern. However, it is important to note that for this experiments, the cells were cultured and differentiated on brain-derived scaffolds, which were subsequently fixed in formalin, processed into paraffin blocks, and sectioned for immunostaining. This process, while necessary for maintaining scaffold structure and enabling sectioning, may result in partial cell shrinkage or distortion of cytoplasmic structures, which could affect the apparent morphology and distribution of cytoplasmic markers, such as TUJ1 and TH proteins. This may partly explain the limited neuronal morphology and nuclear-like signal observed in the current images. Confocal microscopy is the preferred technique in this context; however, it exceeded the scope of our project and available resources.

Unfortunately, due to unforeseen circumstances beyond our control, we could not repeat the immunocytochemistry experiments. However, we have adjusted the contrast and brightness of the images in the revised figure to address the visualization concerns to some extent. Although we were unable to perform quantitative analysis of TH and TUJ1 expression, the results from gene expression (qPCR) and protein expression (western blot) collectively support the neuronal and dopaminergic differentiation of the ADSCs. Finally, it is worth noting that in our earlier research, using the same induction medium, we assessed the functional maturation of dopaminergic neurons generated from hADSC in 2D culture on gelatin-coated plates and validated through HPLC analysis that a considerable amount of dopamine was released in response to KCl-induced depolarization [10].

Thirdly, the control for the decellularized rat brain slices is not appropriate, or at least, other more appropriate controls should have been included. Why gelatin for example - this is not used in standard cell culture differentiation protocols and is not an ECM component highly expressed in brain tissue.

Response: We agree with the reviewer that gelatin is not an ECM component highly expressed in the brain tissue. The primary objective of our study was not to compare our decellularized brain ECM with other 3-dimentional systems or various decellularized tissue ECMs. Instead, we aimed to develop a robust protocol for the decellularization of brain tissue. In our preliminary investigation, we examined multiple concentrations and treatment durations for sodium deoxycholate, Triton X-100, and trypsin/EDTA, alongside varying shaker incubator speeds. We found that the methodology outlined in this paper effectively removes the cells and genetic materials while retaining the ultrastructure and composition of the ECM throughout the decellularization process. Furthermore, we aimed to assess the dopaminergic differentiation of ADSCs on the brain ECM, as a 3-dimentional system, in comparison to our previously established two-dimensional system. Gelatin was selected as a coating material solely to enhance cell adhesion, facilitate cell spreading, and support cell proliferation and viability. It is noteworthy that gelatin-coated substrates are extensively used in cell culture applications for fibroblasts, neurons, and stem cells.

Finally, the data cannot be compared to existing literature as the time point for differentiation was not specified.

Response: Thank you for your helpful comment. We acknowledge the omission of the total differentiation period in the original manuscript. This has now been incorporated in the revised version, where we have specified that the differentiation was performed over 12 days. We appreciate your attention to this detail.

Reviewer #4:

This is a general good original research. The authors must address these questions:

1. what are the main novelties of this research? It seems that a new method for decellularization has been conducted, but its efficacy and effectiveness has not been compared with common traditional approaches.

Response: The main novelty of this research was firstly to establish a decellularization method for brain tissue that efficiently removes the cells and genetic material while retaining the ultrastructure and composition of the ECM during the decellularization process. Our goal in this project was not to compare our method with previously published ones, although it would have been better to do so.

Secondly, we aimed to assess the dopaminergic differentiation of human ADSCs on this decellularized brain ECM. As mentioned in the introduction “Previous studies have demonstrated the importance of using decellularized brain ECM to improve differentiation and maturation of neurons, with various cell sources such as neuroblastoma cells [11], induced pluripotent stem cell (iPSC)-derived neurons [12], neural stem cells [13, 14], and bone marrow-derived mesenchymal stem cells (BM-MSCs) [15] being utilized as the initial cells. However, no study has specifically examined the influence of brain ECM on the dopaminergic differentiation of stem cells in vitro”.

2- Since the control group of cells has received dopaminergic culture medium in two dimensional flask condition, the test group was cultured on 3D bio-scaffold. As that are substantial differences between two and three dimensional culture condition, the elevation of gene expression could not be exclusively related to the de-cellularized bio-scaffolds. More intricate and complex controlled condition is necessary to draw the claimed conclusions.

Response: As mentioned in the answer to the previous question, in this project, we aimed to establish an efficient decellularization method for brain tissue and to examine dopaminergic differentiation of human ADSCs on it. Since we previously set up dopaminergic differentiation of human ADSCs in 2-dimensional culture in gelatin-coated plates, we used this system just to show by gene and protein analysis that dopaminergic differentiation on decellularized brain ECM is efficient. Albeit we agree with the reviewer that in the next step, we should compare our decellularization and differentiation methods with a previously established 3D culture system.

3- did the authors conduct any essay for cell sorting? Extraction of human adipose derived mesenchymal stem cells usually leads to a heterogeneous colony of the cells and scientific sorting approaches is necessary for compromising a homogeneous colony. Otherwise the results seem not to be repetitive.

Response: As shown previously, ADSCs do not consistently express all characteristic MSC markers immediately after isolation, but some specific surface markers (e.g., CD105, CD166) increase during culture, while the expression of others decreases (e.g., CD34). During culture, the heterogeneity of ADSCs decreases, and it seems that the characteristic marker expression of ADSCs depends on the culture conditions and time in culture. ASCs in passages 2 or 3 are morphologically a homogeneous population of fibroblastoid cells. These cells uniformly express the characteristic markers of MSCs, including CD29, CD44, CD73, CD90, CD105, and CD166, and lack the expression of CD11b, CD14, CD31, and HLA-DR [16].

In our study, we assessed the third-passaged ADSCs for the expression of MSC markers by flow cytometry (Fig 1) and showed that more than 98% of the cells were positively stained with the antibodies against CD105, CD73 and CD90 proteins, while Only 0.3% of the cells showed expression of CD45 as a hematopoietic marker.

4- For further and more complete investigation it is needed to assess the expression of neuron exclusive genes in both mRNA and protein levels.

Response: Thank you for the valuable comment. We examined the expression of neuron-specific genes, NSE and NEFL, at the mRNA level, and the results have been added to Figure 5. Additionally, neuron-specific proteins, TUJ1 and NEFL, have been previously assessed at the protein level, and their results are shown in Figure 7.

Reviewer 5#

Recommendation: Minor Revision. The manuscript generally meets PLOS ONE’s publication criteria in terms of scientific rigor, completeness of data, and ethical standards. The study is well designed and yields interesting results about using a decellularized rat brain extracellular matrix (ECM) to enhance dopaminergic differentiation of human adipose-derived stem cells (hADSCs). However, a few minor revisions are needed to improve clarity, consistency, and completeness. Below, I provide detailed comments on specific aspects:

Scientific Validity and Methodological Rigor

The experimental design is solid and appropriate for the research question. The authors compare hADSC differentiation on 3D decellularized brain scaffolds versus a standard 2D culture, which is a clear and logical way to test the influence of the ECM microenvironment. Strengths of the methodology include a thorough decellularization protocol (with chemical, enzymatic, and physical steps) and comprehensive validation of decellularization success (DAPI nuclear staining, H&E and Masson’s trichrome histology, DNA quantification, laminin immunostaining, and SEM imaging). These confirm that cellular material was effectively removed while preserving ECM structure – lending credibility to the scaffold’s integrity. The hADSCs are well-characterized prior to use (positive for MSC markers CD105, CD73, CD90 and able to undergo adipogenic/osteogenic differentiation), ensuring the starting cell population is appropriate and multipotent. The differentiation protocol is detailed and uses known dopaminergic-inductive factors (Neurobasal medium with SHH, FGF8, bFGF, later adding BDNF), which is scientifically sound. Multiple outcome measures are employed – including RT-PCR/qPCR for gene expression, immunocytochemistry/immunohistochemistry for neuronal markers, and Western blotting for protein levels – providing convergent evidence to support the conclusions. Overall, the study’s methods and analyses are rigorous and support the validity of the findings.

Areas for improvement: A few clarifications could further strengthen the rigor.

1. the authors should quantify some of the qualitative observations. For example, immunostaining revealed a “significant proportion” of cells positive for TH and TUJ1 in both conditions – it would improve the paper to quantify the percentage of cells expressing these markers on the scaffold vs. on 2D culture. This would give the reader a clearer idea of differentiation efficiency and how much the 3D ECM improves outcomes (e.g. “X% of cells were TH-positive on ECM vs. Y% on 2D”). If such quantification was done, it should be reported; if not, even a rough estimate or qualitative statement in the Discussion noting the difference in positive cell abundance would be helpful.

Response: Thank you for your valuable feedback. We agree that quantifying the percentage of cells expressing dopaminergic-specific markers would strengthen the manuscript and provide readers a clearer idea of differentiation efficiency. However, the potential infiltration of certain cells into the scaffold, along with the fragility of neurons, presents challenges in isolating the cells from the scaffold and accurately quantifying their percentage. As a result, we were unable to provide exact percentages for TH-positive cells on the scaffold compared to those in 2D culture. It is worth noting that previous studies documenting neural differentiation on decellularized scaffolds also did not specify such percentages. Nevertheless, in our previously published research utilizing the same induction medium, we demonstrated through flow cytometry analysis that approximately 60% of hADSCs expressed TH following differentiation in 2D culture.

We hope this explanation clarifies our limitations and the context of our findings. We will consider including a brief statement in the discussion to offer a rough indication of differentiation efficiency within our system.

2. While the upregulation of dopaminergic genes (e.g. TH, GLI1, VM

---

## [Editor Report · Decision Letter 2]

Decellularized rat brain extracellular matrix effectively induces the dopaminergic differentiation of human adipose-derived stem cells

PONE-D-24-59238R2

Dear Dr. Taha,

We’re pleased to inform you that your manuscript has been judged scientifically suitable for publication and will be formally accepted for publication once it meets all outstanding technical requirements.

Kind regards,

Panayiotis Maghsoudlou

Academic Editor

PLOS ONE
---

## [Editor Report · Acceptance letter]

PONE-D-24-59238R2

PLOS ONE

Dear Dr. Taha,

I'm pleased to inform you that your manuscript has been deemed suitable for publication in PLOS ONE. Congratulations! Your manuscript is now being handed over to our production team.

Kind regards,

on behalf of

Dr. Panayiotis Maghsoudlou

Academic Editor

PLOS ONE